# Health economic evaluation of a serum-based blood test for brain tumour diagnosis: exploration of two clinical scenarios

Ewan Gray,[1] Holly J Butler,[2,3] Ruth Board,[4] Paul M Brennan,[5]
Anthony J Chalmers,[6,7] Timothy Dawson,[8] John Goodden,[9] Willie Hamilton,[10]
Mark G Hegarty,[2,3] Allan James,[11] Michael D Jenkinson,[12,13] David Kernick,[14]
Elvira Lekka,[8] Laurent J Livermore,[15] Samantha J Mills,[12] Kevin O'Neill,[16]
David S Palmer,[3,17] Babar Vaqas,[16] Matthew J Baker[2,3]

EG and HJB contributed equally.

For numbered affiliations see end of article.

**Correspondence to**
Dr Matthew J Baker;
matthew.baker@strath.ac.uk

## ABSTRACT

**Objectives** To determine the potential costs and health benefits of a serum-based spectroscopic triage tool for brain tumours, which could be developed to reduce diagnostic delays in the current clinical pathway.

**Design** A model-based health pre-trial economic assessment. Decision tree models were constructed based on simplified diagnostic pathways. Models were populated with parameters identified from rapid reviews of the literature and clinical expert opinion.

**Setting** Explored as a test in both primary and secondary care (neuroimaging) in the UK health service, as well as application to the USA.

**Participants** Calculations based on an initial cohort of 10 000 patients. In primary care, it is estimated that the volume of tests would approach 75 000 per annum. The volume of tests in secondary care is estimated at 53 000 per annum.

**Main outcome measures** The primary outcome measure was quality-adjusted life-years (QALY), which were employed to derive incremental cost-effectiveness ratios (ICER) in a cost-effectiveness analysis.

**Results** Results indicate that using a blood-based spectroscopic test in both scenarios has the potential to be highly cost-effective in a health technology assessment agency decision-making process, as ICERs were well below standard threshold values of £20 000–£30 000 per QALY. This test may be cost-effective in both scenarios with test sensitivities and specificities as low as 80%; however, the price of the test would need to be lower (less than approximately £40).

**Conclusion** Use of this test as triage tool in primary care has the potential to be both more effective and cost saving for the health service. In secondary care, this test would also be deemed more effective than the current diagnostic pathway.

## INTRODUCTION

At an average of 20 years, patients with malignant brain tumours have the highest number of years of life lost, compared with all other primary cancers.[1] This, at least in part, may relate to diagnostic delays, reflecting the non-specific early symptoms, such as headache and dizziness, from which general practitioners (GP) must identify patients at risk for further investigation. The lack of a low-cost diagnostic and/or screening tools available within the health service contributes to this delay. We have recently demonstrated that a spectroscopic test using blood serum is able to effectively identify brain tumours in patients with sensitivities and specificities as high as 92.8% and 91.5%, respectively, in a

**Strengths and limitations of this study**

► Simplified models of clinical pathways were mapped with input from, and consensus among, a wide range of clinical experts including neurosurgeons, neuro-oncologists, neuropathologists, neuroradiologists and primary care experts.

► The spectroscopic blood test was highly sensitive and specific in retrospective data, with performances of 92.8% and 91.5%, respectively. There is potential for this to contribute towards improved prognosis for patients, as well as healthcare savings.

► This study is based on proof-of-concept studies, in advance of a pending prospective clinical trial. As these samples are retrospective, there is the possibility the diagnostic performance will not be as high in prospective studies.

► A lack of clinical trial evidence necessitates the estimate of long-term benefits of improved diagnostic protocols based on disease natural history models. This creates additional uncertainties.

► The precise patient population for whom the test may be suitable in the primary care setting is difficult to establish at this stage in development. This study considers a limited definition of eligibility that may need revision in light of future evidence.

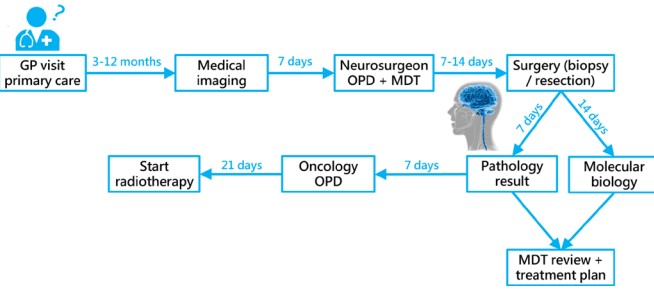

**Figure 1** The diagnostic pathway of brain tumours. Timings relate to the diagnosis of high-grade gliomas and are based on discussion with the Clinical Focus Team and Aggarwal *et al*.[5] GP, general practitioner; MDT, multidisciplinary team; OPD, outpatient department.

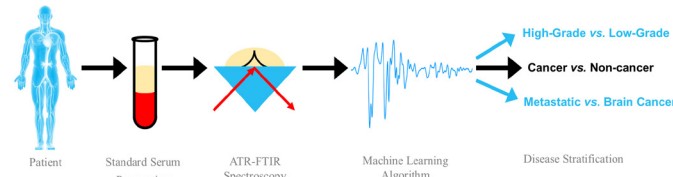

**Figure 2** Attenuated total reflectance Fourier-transform infrared (ATR-FTIR) spectroscopic test of blood serum for the diagnosis and stratification of brain tumours using machine learning algorithms.

tissue bank case–control series.[2 3] This approach is based on Fourier-transform infrared (FTIR) spectroscopy and can detect disease-specific signatures, which are extracted mathematically using pattern recognition and machine learning algorithms.

## Current diagnostic pathway

Currently, patients who are symptomatic with a brain tumour visit their GP on average five times before being referred to secondary care.[4] Partly as a result of this diagnostic delay, up to 61% of brain tumour diagnoses occur in an emergency setting, often following a seizure.[5] Patients diagnosed by the emergency route have a poorer prognosis.[6 7] For some patients this may be because the disease is at a more advanced stage at diagnosis. The complications precipitating the emergency admission make an additional contribution to mortality.

Screening programmes for breast, prostate, cervical and colorectal cancers have proved effective for diagnosing patients at an earlier stage, which can result in a better prognosis.[8–11] These screening programmes have had a significant impact in reducing the number of patients presenting as an emergency. To date there has been no accessible and economically viable diagnostic tool for early detection of asymptomatic and symptomatic brain tumours.[12] The addition of a rapid and accurate blood test for patients with suspected primary brain tumours (symptomatic patients) therefore has the potential to improve outcomes by allowing prioritisation of patients most at risk of a brain tumour for further investigation. Under the current patient pathway it is not feasible to provide fast-track diagnostic imaging because the number of patients with non-specific headache symptoms is very large and the positive predictive value (PPV) on the basis of symptoms alone is less than 3% for all symptoms other than a new-onset seizure.[13 14]

MRI and CT are the current gold standard for identifying structural brain lesions including tumours. Treatment decisions made at the neuro-oncology multidisciplinary team meeting are often based on the imaging alone. Following surgical resection or biopsy with histopathology and molecular analysis, definitive treatment can be planned.[15] Surgery to secure the tissue diagnosis has

a small risk to the patient of neurological deterioration and death.[16]

The diagnostic pathway also represents a significant cost burden to the health service, with a single MRI and CT scan in the UK costing around £164 and £85, respectively (National Schedule of Reference Costs (2014–2015), Medicare Physician Fee Schedule 2016). A typical timeframe of the diagnostic pathway for brain tumours, specifically primary gliomas, is illustrated in figure 1, and effectively highlights the significant wait that a symptomatic patient may have before receiving brain imaging. Even from this stage, regardless of the time to GP referral or emergency presentation, full diagnosis may take a further 5 weeks.

## Serum spectroscopic diagnostics

This novel blood test for early brain tumour detection is based on the interaction of infrared (IR) light with biological components of blood serum (figure 2). Specifically, using attenuated total reflectance (ATR)-FTIR spectroscopy, specific bond vibrations of given molecules can be elucidated from serum samples, thus providing a unique insight into the composition via an absorbance spectrum.[17] Benefits of an ATR-FTIR-based approach include a robust, user-friendly methodology without extensive sample preparation, which would readily fit into a clinical setting.[18] In short, serum is obtained according to standard protocols and can be snap frozen and stored at −80° until the point of analysis. A small volume of serum is required for analysis (1–5 μL), which is pipetted onto a crystal, known as an internal reflection element, where IR non-destructively interacts with the sample and produces an IR spectrum, with peaks representative of known bond vibrations and hence biomolecular constituents.

Blood serum is a complex medium that contains a variety of biomolecules, including around 20 000 proteins, which may be employed as diagnostic biomarkers.[19] In the case of brain tumours, such blood-based technologies are limited, due to a lack of an established brain tumour-specific diagnostic biomarker.[20] With our spectroscopic approach, rather than derive single biomarker-specific information, a global signature is obtained which encompasses the entire biomolecular makeup. This is epitomised as an equally complex biological absorbance spectrum, which contains a wealth of diagnostic information (an example may be seen in figure 3).

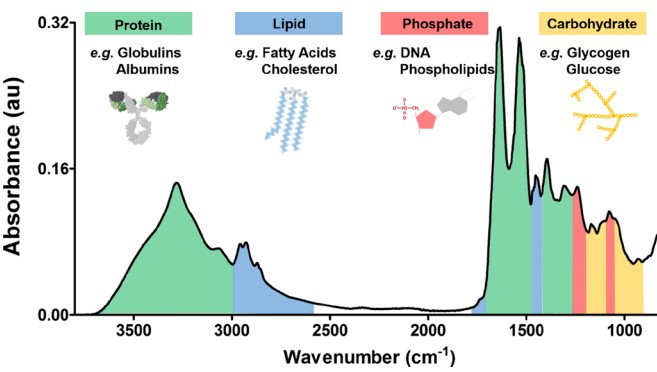

**Figure 3** An unprocessed spectrum derived from human blood serum using attenuated total reflectance Fourier-transform infrared (ATR-FTIR) spectroscopy. Spectral regions correspond to known bond vibrations and can therefore be associated with groups of biomolecules such as protein, lipid, phosphate and carbohydrates. Broad examples of blood serum constituents are listed.

Pattern recognition and machine learning algorithms using spectral features from FTIR data have been demonstrated as rapid and accurate for separating primary brain cancer and non-cancer cases.[2 21] When these algorithms are applied to this information rich data set, the relationship between all biological components of the sample is ascertained, providing a multidimensional analysis of the sample.

In the case–control setting, this approach has been able to detect between cancer and non-cancer, and stratify based on cancer pathology.[2] For further information and in-depth description of the methodology, we direct the readers to the following fundamental review and recent research papers.[2 3 17 21] The ability to triage patients likely to have a brain tumour based on serum sample alone raises the possibility of systematic triaging prior to investigation with more expensive (MRI/CT imaging) and invasive (biopsy) tests. One major impact of having a serum test available would be a possible reduction in the number of unnecessary brain scans; however, as this test is also able to differentiate between primary and secondary tumours, there could also be a knock-on reduction of chest and abdomen scans which are conducted to rule out primary disease elsewhere. There is also the possibility that this approach will reduce the incidence of incidental abnormalities which in themselves can cause considerable distress. Ultimately, it is expected that this could allow earlier and potentially more effective treatment of brain tumours.

It is important to note that this study is based on proof-of-concept studies, in advance of a pending clinical trial. As these samples are retrospective, there is the possibility the diagnostic performance will not be as high in prospective studies. In addition to determining the true diagnostic accuracy of the technique, the planned clinical trial held in primary care will also reveal the suitable patient population for the test, as well as the long-term benefits of an improved diagnostic pathway.

## Aims and objectives

The aim of the economic evaluation is to assess the potential cost-effectiveness of this spectroscopic technology, in advance of any prospective study results being available. There are three main objectives for the evaluation; first to create a map of where the test could be used in the clinical pathway. The second is to assess the potential cost-effectiveness of the technology, if the performance shown in the case–control study is replicated prospectively. This will give an indication of whether the technology would meet the criteria for acceptance for use in the National Health Service (NHS) that are applied by health technology assessment (HTA) agencies. Related to this, the third objective is to define the level of performance in prospective trials, and any additional evidence that would be needed to meet the cost-effectiveness criteria of an HTA decision-making process. This can include diagnostic performance and also effects on long-term outcomes such as survival and resource use. To achieve these objectives a simplified economic model of two important clinical scenarios is used to explore cost-effectiveness.

## METHODOLOGY

### Mapping the clinical pathway

In order to appreciate the current clinical pathway and determine an appropriate entry point for a serum spectroscopic test, a pan-UK clinical focus group was established. This cohort included neurosurgeons, clinical and medical oncologists, neuropathologists, neuroradiologists, academic GPs with special interests and experts in primary care diagnostics (see online supplementary appendix 1).

### Cost-effectiveness analysis

A cost-effectiveness analysis was conducted to calculate the effects on health outcomes and health service costs of introducing spectroscopic testing in each of two scenarios. The health outcomes considered were life-years and quality-adjusted life-years (QALY).

A decision tree was used to model the pathway for patients presenting with symptoms warranting a referral for MRI/CT imaging for suspected brain tumour (figure 4). Separate models were considered for primary and secondary care. The time horizon of the model is 2 years and the perspective is that of the healthcare service. A 2-year horizon was selected because of the short duration of survival in this patient group: median survival is approximately 1 year for high-grade gliomas, which are the most common malignant primary brain tumour. In all scenarios the comparator is the current diagnostic pathway (ie, imaging alone). Further details regarding the node probabilities, simplifications and assumptions of the model can be found in online supplementary appendix 2.

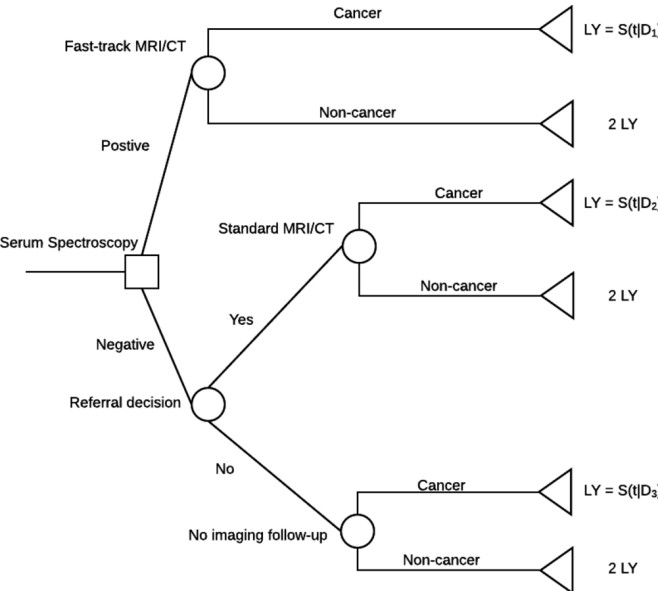

**Figure 4** A decision tree model describing the integration of a serum spectroscopy test in the current diagnostic pathway, and the effect on MRI/CT imaging for suspected brain tumour. D1, 1 week; D2, 4 weeks; D3, 8 weeks; LY, life-year; S(t|D), survival time in days conditional on 'delay'.

## Diagnostic performance

The sensitivity and specificity of the test for detecting brain tumours has been demonstrated in a series of case–control studies using historical samples from the Brain Tumour North West and Walton Centre NHS biobanks.[2 3] In the key case–control study nine FTIR spectra were collected from each of 433 patients.[3] Of these, 134 were from patients with primary brain tumours (64 high-grade glioma), 177 were from patients with cerebral metastases and 122 were from non-cancer controls. FTIR spectra were analysed using the random forest method to fit a classification model. Classification performance was estimated by applying the fitted model to a test set containing 20% of the patients from the original data set that were not used in the model fitting step (hold-out test set). Classification statistics are computed as averages of this process, iterated 96 times using random training and test sets. Under the best available model, sensitivity estimated by this method was 92.8% and specificity was 91.5% for the analysis of cancerous versus non-cancerous serum.[3] These classification statistics were established using 96 independent iterations of a random forest model, and resulted in SDs of 1.1% and 1.9%, respectively.

Establishing whether the performance demonstrated case–control data from historical samples translates to equivalent performance when applied prospectively in clinical practice is the subject of a planned clinical trial. This is critical to demonstrating the clinical and cost-effectiveness of a serum spectroscopy test as part of the diagnostic pathway for brain tumours.

## Prevalence of disease

Prevalence data were sourced from the literature based on clinical expert guidance. Brain tumour prevalence in scenario 2 (secondary care) was assumed to be 3% based on observed rates of primary brain tumour diagnosis among patients referred for brain imaging for suspected cancer in secondary care.[22–24] In scenario 1 (primary care) an estimated prevalence of 0.5% is used based on case–control evidence and expert opinion of the prevalence among patients who would be considered for direct access imaging, using MRI or CT, where this is available and referral to neurology where it is not.[25 26] An alternative prevalence of 1% was explored for scenario 1 based on unpublished data from a direct access imaging service in the UK (P Brennan, personal communication).

The effect of serum spectroscopy testing on the time to diagnosis and time to treatment is discussed in online supplementary appendix 3, in addition to the effect of testing on the use of imaging studies, and also on the patient outcomes. The primary assumptions in this model are that first, the expected time to diagnosis would match the current median time to diagnosis for patients presenting with brain tumour in emergency care as observed in Aggarwal et al's study.[5] Furthermore, based on expert opinion, it is assumed that in secondary care all patients would continue to imaging, while in primary care 50% would continue to imaging following a negative spectroscopy result. This is a conservative estimate associated with the possibility that an imaging test will still be required in some cases based on interpretation of a patient's symptoms and the other non-tumour diagnoses being considered by the clinician. Finally, the effects of early diagnosis on the outcome of brain tumours are estimated using literature describing fitting natural history models to observational data sets of patients with high-grade glioma.[1 27]

## Utility weights

Health state utility weights are applied to life-years to generate QALYs. A systematic review of health state utility weights for high-grade glioma, the most common and aggressive primary brain tumour, was conducted to identify suitable utility weights. Due to heterogeneity it was not considered suitable to pool the estimates. The most appropriate health state utility weight was taken from a previous UK economic evaluation of glioblastoma treatment.[28] A value of 0.89 was used in the base case.

## Resource use and costs

Resource use includes the application of a spectroscopic serum test to all patients prior to imaging, the imaging studies used in the diagnostic process, outpatient neurology clinic visits and GP visits. In the UK analysis, unit costs for imaging studies are taken from UK NHS reference costs (2014/2015), clinic and GP visits from the Personal Social Services Research Unit (PSSRU) costs schedule (table 1). In the US analysis, unit costs are taken from Medicare reimbursement schedules. Unit costs of

**Table 1** Unit costs and comparison for the brain tumour diagnostic pathway

| Item | UK cost per unit (£) (2015 prices) | USA cost per unit ($) (2016 prices) | Source(s) |
|---|---|---|---|
| CT imaging study (CT head) | 85 | 163 | National Schedule of Reference Costs (2014–2015), Medicare Physician Fee Schedule (2016) |
| MRI imaging study (MRI brain with contrast) | 164 | 380 | National Schedule of Reference Costs (2014–2015), Medicare Physician Fee Schedule (2016) |
| Neurology outpatient appointment | 35 | 76 | PSSRU (2016), Medicare Physician Fee Schedule (2016) |
| GP visit | 47.25 | 76 | PSSRU (2016), Medicare Physician Fee Schedule (2016) |
| Stable disease monitoring costs | 116 per 3 months | 154 | [28], exchange rate adjusted |
| Serum spectroscopy test | Lower limit: 50 Upper limit: 100 | Lower limit: 100 Upper limit: 200 | Assumed prices |

GP, general practitioner; PSSRU, Personal Social Services Research Unit.

the test were applied at an upper bound and lower bound rather than a single value as these products have not yet been commercialised. Bounds were set by consultation with scientists developing the tests. Additional resource use and cost assumptions are described in online supplementary appendix 3.

### Incremental cost-effectiveness ratios

The comparative cost-effectiveness of spectroscopic testing compared with no testing is summarised by the incremental cost-effectiveness ratio (ICER) defined as:

$$ICER = \frac{C_s - C_n}{H_s - H_n}$$

$C_s$ and $C_n$ are the total costs with spectroscopic testing and no testing, respectively. Equivalently, $H_s$ and $H_n$ are the total QALYs with and without spectroscopic testing. The ICER can be interpreted as the additional cost per QALY gained.

### Base case and sensitivity analysis

ICERs were calculated for scenarios 1 and 2 using the base case parameter estimates. Base case analysis was repeated for UK and USA settings. Additional sensitivity analyses are reported for the UK setting only.

Sensitivity analyses included one-way sensitivity analysis (OWSA), systematically varying a single parameter in the model, and scenario analysis in which specific model assumptions were altered. OWSA was conducted for test sensitivity, specificity and test cost. Scenario analyses included assuming an additional consultation cost for discussion of test results, assuming a higher proportion of patients continue to imaging following a negative spectroscopy result in primary care and using mean survival rather than median survival.

A probabilistic sensitivity analysis (PSA) was conducted to explore the effects of joint uncertainty in the parameter estimates on the model results.[29]

### Patient and public involvement

Patients and the public were not actively involved in the formation of this study. The impact of the test on clinical decision-making was the priority in this instance; however, the involvement of patients going forward will be fundamental to understanding the tests uptake into the health services.

## RESULTS
### Mapping the clinical pathway

For the first study objective, initial discussion with clinical experts indicated that there are potential uses for the test in both primary and secondary care. The main advantage of employing a cost-effective spectroscopic blood test in the diagnostic pathway is to use it as a triage test. This prioritises more urgent cases for access to services, and acts as a gatekeeper (requiring a positive result in some conditions to give access to services) for imaging studies.

Two clinical scenarios are mapped out below and subsequently explored in this early economic evaluation of the serum spectroscopy test for aiding diagnosis of brain tumours.

### Triage tool in primary care

The primary care scenario explores a population of patients with a clinical presentation that warrants further investigation of possible brain tumour. This would include some patients with headaches and some with focal neurological deficits. This is the group of patients who would be considered for direct access imaging, using MRI or CT, where this is available and referral to neurology where it is not.[30] The blood test is used to provide rapid information, within 24 hours, where a positive result would lead to patients receiving more timely access to imaging. It may also be the case that negative test results, in addition to establishing the low probability of a brain tumour, could also provide some reassurance for those patients who must wait for imaging. The total volume of tests would be approximately 75 000 per year in the UK (see online supplementary appendix 4 for further details).

## Triage tool in secondary care

In this scenario the population is the group of patients who are currently referred for imaging studies from secondary care for suspected brain tumour, typically via neurology clinics. This is the patient group for whom the clinical presentation has the highest PPV. However, even in this high-risk group, the odds of a brain tumour being present are approximately 1:33.[22–24] Again, the spectroscopy test is used to provide rapid information to allow a subset of these patients to access immediate imaging and provide reassurance to other patients who may have to wait longer for definitive imaging studies and diagnosis. The extent of the benefits of triage in this scenario is likely to vary by locality depending on the capacity constraints on imaging and pathology services. This evaluation uses estimates of the delays in diagnosis, and potential improvements in the speed of diagnosis, from a consecutive patient case series in London, UK.[5] The total volume of tests if this scenario occurred would be approximately 53 000 per year in the UK (see online supplementary appendix 4 for further details).

### Cost-effectiveness assessment

The standard threshold value per QALY gained in the UK is considered to be between £20 000 and £30 000. Below this value, a healthcare intervention may be considered cost-effective, whereas a negative ICER value would be deemed cost saving. Base case results for primary care (scenario 1) and secondary care (scenario 2) are presented in table 2. Note results are reported for cohorts of 10 000 patients.

ICERs were well below standard threshold values of £20 000–£30 000 per QALY gained used in the UK, and similar thresholds used internationally, provided the test cost did not exceed £100. The base case results demonstrate that the serum spectroscopy test dominates (more effective and less costly) standard care at the lower bound of test cost in the primary care setting in both the UK and USA. At the upper bound of test cost the ICERs may be within commonly used thresholds or, in the case

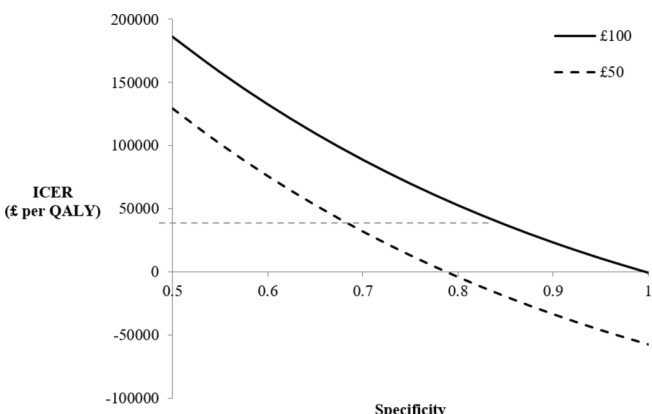

**Figure 5** Incremental cost-effectiveness ratios (ICER) at various specificities in primary care ('Scenario 1'). The £30 000 ICER threshold is displayed as a dashed horizontal line. QALY, quality-adjusted life-year.

of the USA, remain dominant to standard care. In the secondary care setting ICERs of £9982 and $10 153 at the lower bounds of test cost indicate that this test is potentially cost-effective in this setting. At the upper bounds ICERs may still be within commonly used thresholds for cost-effectiveness.

### Sensitivity analysis results

The performance of the test with regard to levels of sensitivity and specificity is addressed using sensitivity analysis. OWSA results for a range of test specificities are displayed in figures 5 and 6, for primary and secondary care, respectively, displaying the ICER with varying test specificity. Note that the estimated QALYs do not change with specificity in the model, therefore changes in the ICER are due solely to changes in incremental costs. Varying sensitivity changes both estimated QALYs and estimated costs therefore results of the OWSA for test sensitivity are presented on the cost-effectiveness plane (online supplementary appendix 5). In primary care, using the upper cost limit of £100 it is evident that the test is deemed cost-effective at specificities of around 0.9 and above, where the ICER

**Table 2** Incremental QALYs, costs and ICERs for scenarios 1 and 2, UK and USA

| Serum spectroscopy test cost (£) | Scenario 1—primary care | | | Scenario 2—secondary care | | |
|---|---|---|---|---|---|---|
| | ΔQALY | ΔCost | ICER | ΔQALY | ΔCost | ICER |
| UK | | | | | | |
| 50 | 8.81 | −422 116 | −47 913 (dominates) | 52.86 | 527 646 | 9982 |
| 100 | 8.81 | 77 884 | 8840 | 52.86 | 1 027 646 | 19 441 |
| USA | | | | | | |
| 100 | 8.81 | −1 718 475 | −195 058 (dominates) | 52.86 | 536 702 | 10 153 |
| 200 | 8.81 | −218 475 | −24 798 (dominates) | 52.86 | 2 036 702 | 38 530 |

ΔQALY, ΔCost: difference is QALYs/costs (with serum spectroscopy test—without test), 10 000 patients.
ICER, incremental cost-effectiveness ratio; QALY, quality-adjusted life-year.

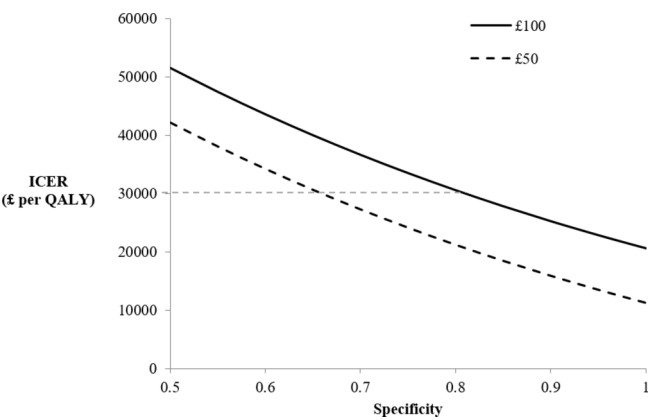

**Figure 6** Incremental cost-effectiveness ratios (ICER) at various specificities in secondary care ('Scenario 2'). The £30 000 ICER threshold is displayed as a dashed horizontal line. QALY, quality-adjusted life-year.

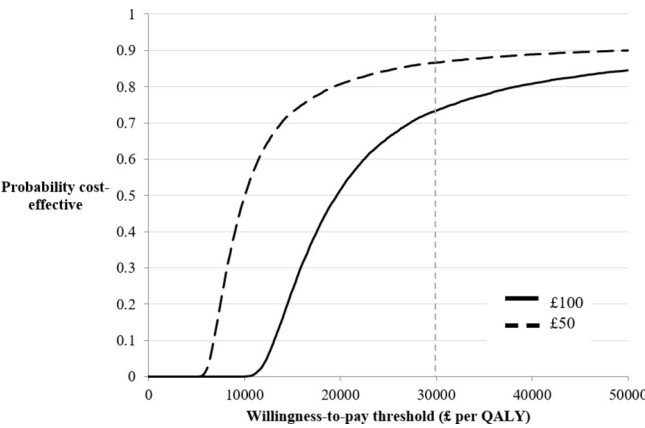

**Figure 7** Cost-effectiveness acceptability curve (CEAC) at £50 and £100 per test—primary care ('Scenario 1'). QALY, quality-adjusted life-year.

is below standard thresholds. In contrast, at the lower cost limit, the test is cost-effective at specificities above 0.8. Although the serum spectroscopy test is not cost saving at low cost (or near perfect specificities), the test is still considered cost-effective at specificity levels around 0.7 and 0.8 for £50 and £100 pricing, respectively.

The OWSA results highlight how ICERs in scenario 2 (secondary care) are strongly influenced by test sensitivity while ICERs in scenario 1 (primary care) are more strongly influenced by test specificity. These features are a result of the varying prevalence of disease and the assumption in scenario 2 that 100% of patients are referred for imaging following negative result. It should be noted that test specificity is important in both scenarios. Relatively small improvements in test specificity can substantially change the ICER, while larger improvements in test sensitivity are required to substantially alter the ICERs.

Additional scenario analyses are also reported in online supplementary appendix 5. These demonstrate that results are robust to using mean survival estimates rather than median survival estimates and including additional consultation costs for positive test results. If the prevalence of brain tumours in scenario 1 is 1% rather than 0.5% the incremental QALYs increase substantially and the ICERs are reduced.

The PSA results reported in figures 7 and 8 indicate that at a test cost of £50 and an ICER threshold of £30 000 per QALY there is a near 100% probability the serum spectroscopy test is cost-effective in scenario 1 and approximately 90% probability it is cost-effective in scenario 2. The corresponding probabilities at the upper bound cost of £100 are approximately 85% and 75%.

## DISCUSSION

This economic evaluation establishes the potential for serum spectroscopy to have a role in the diagnosis of both benign and malignant brain tumours in both primary and secondary care. The potential costs and health benefits of testing using a spectroscopic method prior to CT/MRI

tests (or in some scenarios to avoid imaging) have been estimated based on a mathematical model with parameter values taken from published studies and expert opinion. This diagnostic tool is sensitive to all brain tumours (benign or malignant); however, this assessment is closely aligned with the diagnosis of primary gliomas, where there is a maximum potential benefit to the health service.

The major limitations of this analysis relate to the use of proof-of-concept studies and a disease natural history model rather than direct clinical trial evidence. This creates additional uncertainties. Results should be interpreted as indicative and used primarily to guide future evidence generation. Furthermore, the scenarios explored were limited in scope; future studies should continue to refine understanding of the role of the test in real-world clinical decision-making.

When used as a triage tool in primary care, this novel test has the potential to deliver improvements in health outcomes and also to reduce costs. At the lower end of test costs, the technology would be cost-saving for the health service. At higher test costs the technology is still

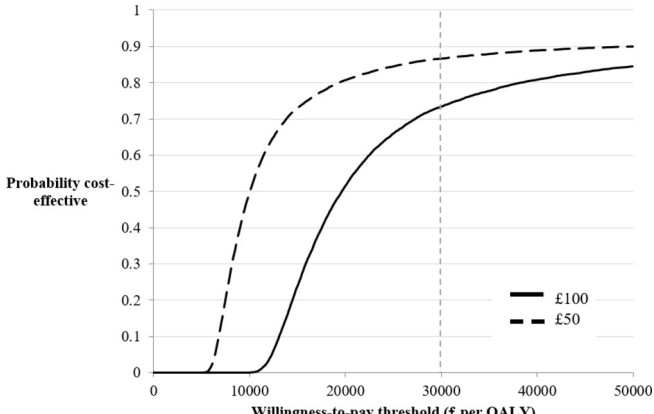

**Figure 8** Cost-effectiveness acceptability curve (CEAC) at £50 and £100 per test—secondary care ('Scenario 2'). QALY, quality-adjusted life-year.

likely to be considered cost-effective in HTA agency decision processes.

In scenario 2, in which serum spectroscopy is used as a triage tool in secondary care, the technology will create additional costs and also produce sizeable health benefits. At test costs of under £100 ($200) the technology would be likely to be considered as a cost-effective use of resources in HTA agency decision processes in the UK (and USA). It is assumed that in both scenarios, the uptake of the test in the USA would mirror that of the UK; however, this would need to be explored further, alongside clinical experts of the USA care pathways.

Sensitivity analyses have demonstrated the importance of diagnostic performance on the cost-effectiveness of the test. In particular, test specificity is important in the primary care setting. If test specificity is 87.5% or worse, the technology may not be considered cost-effective at higher values of assumed test cost. This is due to the increased number of false positive results in this low prevalence population, generating a greatly increased proportion of fast-track imaging studies which increases costs.

To strengthen the case that this approach represents a cost-effective use of healthcare resources it is necessary to establish the diagnostic performance of the test prospectively. This can be accomplished by a suitably large cohort study in which serum spectroscopy is used alongside current clinical practice in one of the patient groups included in this evaluation. It would be appropriate to initially target the secondary care patient population, because the higher prevalence of disease in this group will reduce the sample size needed to accurately estimate diagnostic performance.

Decision makers are often most interested in patient outcomes, such as survival, rather than intermediate outcomes, such as accuracy or speed of diagnosis (although this latter point is vital for treatment of high-grade gliomas). From this perspective, a randomised trial, or a prospective cohort study with extended follow-up, may be required to fully establish the size of survival and quality-of-life benefits of including a serum spectroscopy test in the diagnostic pathway. A trial with primary outcomes relating to survival and quality of life would be specific to either the primary or secondary care setting (rather than generalisable to both), would need a large sample size and would also require a follow-up period to capture survival benefits. In the case of malignant glioma this would require a period of at least 24 months. Such a trial would clearly be expensive, time consuming and may be unfeasible. Decision makers may be willing to make a decision on implementation of the blood test based on the modelled effects of improvements in intermediate outcomes on later patient outcomes. In this situation, the model proposed in this evaluation, populated with diagnostic performance and other data from a prospective trial, could be used to inform decisions about the wider adoption of the technology.

Future developments beyond trials such as emerging epidemiological evidence and new technologies should also be included in any future evaluations. It was not possible to foresee and include all such possible scenarios in this early evaluation but that should not preclude assessment in the light of new evidence. Updated analysis should inform any decisions about system-wide implementation.

Several results in this analysis suggest cost savings through reduced use of imaging for patients with a negative test result. To make the case that a serum spectroscopic test can improve the efficiency of the diagnostic pathway prospective studies will also need to explore the impact of these test results on clinician and patient imaging study decisions. The possibility remains that the test may triage patients, but may not reduce the number of scans being conducted, and could potentially increase the demand on imaging. For example, if the test is applied to a wider population than intended in primary care due to the availability of such a non-invasive test effectively lowering the threshold for investigation. This highlights the need to study decision-making in this area prior to any implementation in primary care. Nevertheless, this triaging of patients would still benefit each patient who is provided with an early diagnosis.

This evaluation has explored the potential for serum spectroscopy to be a cost-effective addition to the diagnostic pathway for brain tumours. It has demonstrated that in specific scenarios this novel test may be an effective and cost-effective technology in reducing the delay to diagnosis for patients with brain tumours. Prospective trials are required to provide definitive evidence.

**Author affiliations**
[1]Health Improvement Scotland, Glasgow, UK
[2]Department of Pure and Applied Chemistry, University of Strathclyde Technology and Innovation Centre, Glasgow, UK
[3]ClinSpec Diagnostics Limited, University of Strathlcyde, Technology and Innovation Centre, Glasgow, UK
[4]Rosemere Cancer Centre, Lancashire Teaching Hospitals NHS Trust, Royal Preston Hospital, Preston, UK
[5]Department of Clinical Neurosciences, Western General Hospital, Edinburgh, UK
[6]Beatson West of Scotland Cancer Centre, Glasgow, UK
[7]Institute of Cancer Sciences, Wolfson Wohl Cancer Research Centre, University of Glasgow, Glasgow, UK
[8]Neurosurgery Department, Lancashire Teaching Hospitals NHS Trust, Royal Preston Hospital, Preston, UK
[9]Neurosurgery Department, Leeds General Infirmary, Leeds, UK
[10]Primary Care Diagnostics, University of Exeter Medical School, College House, University of Exeter, Exeter, UK
[11]Institute of Molecular Cell and Systems Biology, Glasgow, UK
[12]Neurosurgery, The Walton Centre NHS Foundation Trust, Liverpool, UK
[13]Institute of Translational Medicine, Clinical Science Centre, University of Liverpool, Liverpool, UK
[14]St Thomas' Medical Group, Exeter, UK
[15]Department of Neurosurgery, Oxford University Hospitals NHS Foundation Trust, John Radcliffe Hospital, Oxford, UK
[16]John Fulcher Neuro-Oncology Laboratory, Imperial College, London, UK
[17]WestCHEM, Department of Pure and Applied Chemistry, University of Strathclyde, Glasgow, UK

**Acknowledgements** The authors acknowledge the support from Rosemere Cancer Foundation, Brain Tumour North West, EPSRC, and the Sydney Driscoll Neuroscience Foundation for funding.

**Contributors** EG conducted the health economic assessment including data analysis, data interpretation and writing. HJB contributed to data interpretation, writing and providing details regarding clinical spectroscopy. MJB is the principal investigator of this research, and initiated this interdisciplinary project and team. RB, PMB, AJC, TD, JG, WH, AJ, MDJ, DK, EL, LJL, SJM, KO and BV are members of the clinical focus group involved in spearheading translation, all of whom provided feedback on the manuscript. MGH, DSP, HJB and MJB are involved in the ClinSpec Diagnostics project.

**Funding** Scottish Enterprise is Scotland's main economic development agency and non-departmental public body of the Scottish Government. They work with partners in both the private and public sectors to exploit the best opportunities to deliver a significant, lasting effect on the Scottish economy. The High-Growth Spinout Programme helps researchers take their ideas and inventions from the lab to the global marketplace. (UKCG 12448)

**Competing interests** MJB, MGH, HJB and DSP are all involved in ClinSpec Diagnostics, a prospective spin-out company from the University of Strathclyde focusing on the translation of serum spectroscopic diagnostics (Company No SC535447).

**Patient consent** Not required.

**Provenance and peer review** Not commissioned; externally peer reviewed.

**Data sharing statement** No additional data available.

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
