## [Reviewer comments · BMJ Open]

ARTICLE DETAILS

TITLE (PROVISIONAL)	Health economic evaluation of a serum based blood test for brain tumour diagnosis: Exploration of two clinical scenarios.
AUTHORS	Gray, Ewan; Butler, Holly; Board, Ruth; Brennan, Paul; Chalmers, Anthony; Dawson, Timothy; Gooden, John; Hamilton, Willie; Hegarty, Mark; James, Allan; Jenkinson, Michael; Kernick, David; Lekka, Elvira; Livermore, Laurent; Mills, Samantha; O'Neill, Kevin; Palmer, David; Vaqas, Babar; Baker, Matthew

VERSION 1 – REVIEW

REVIEWER	Giuseppe Bellisola Department Pathology and Diagnostics, Azienda Ospedaliera Universitaria Integrata Verona, Italy
REVIEW RETURNED	15-May-2017

GENERAL COMMENTS	Based on the data obtained in a previous case-control study (Refs 2 and3), authors built models to explore the potential economic impact of adding a serum blood-based spectroscopic test in the triage for brain tumour diagnosis. They aimed at achieving three main objectives: 1) define where the test could be used in the clinical pathway; 2) assess the potential cost-effectiveness of the technology; 3) define the level of performance necessary to meet the cost-effectiveness criteria of an health technology assessment (HTA) agency necessary to propose of extending trials on large cohorts of patients. Parameters have been carefully selected as well as methodology rigorously applied to build predictive models and obtain health economic evaluation. Results have been extensively illustrated and discussed. Check the correspondence with Figures number (Pages 28-29/40): ICER with varying test specificity is illustrated in Figures 5 and 6 and PSA in Figures 7 and 8, respectively.
---

REVIEWER	W.H. van Harten Netherlands Cancer Institute Amsterdam NL University of Twente, Enschede NL
REVIEW RETURNED	08-Oct-2017

GENERAL COMMENTS	This paper describes the Cost Effectiveness Analysis (CEA) of adding a protein spectrum test in serum to assess the chance of brain tumor presence in 2 scenarios. The literature and background material on the test is not very extensive, so it is very difficult to assess whether the test is really specific for brain tumors (especially the hypothesis that the test
--

covers all brain tumor types is surprising). Further the background publication in J Neurooncol. suggests that the test also discerns other tumor types than brain tumors. I am no specialist in spectroscopy, but remember that early test series in my own institution, revealed very unspecific patterns, that could hardly discriminate between tumors, but also with a range of other diseases. The readers should be reassured in text that there is sufficient evidence that the test is specific for brain tumors and no other (non malignant) diseases are "hit" with this serum protein pattern. It is not described what the findings are of applying this test in a more general headache population. That is important as this may influence the methodological characteristics of the test and especially for the 1st scenario as GP's may send much more patients than expected once the test is commercialized.

Further the authors seem not to be very convinced of the diagnostic value of the test, in view of not only presuming that 50% of the population that is at risk in General Practice, but also all referred patients will still undergo CT/MRI. It would be wise to explain the need for this more extensively, and especially the second scenario seems rather illogical, why adding a spectroscopy test while all patients go through imaging, which is necessary for proper diagnosis anyway. I would expect a scenario in which a proportion of referred pts scoring negatively on the test, does not need imaging. What is the rationale for adding the test in the present scenario?

In early stages of test/marker development those directly involved can be rather optimistic about the value and seem often certain about the actual use in practice. After more thorough research this does not always hold and sometimes indications for application change and even competing tests can appear on which key opinion leaders perspectives may differ thus hampering system wide implementation. None of these options is explored in the discussion; I would consider it unlikely that the test on which only a few publications exist and is explored through CEA in a theoretical population in 2 scenario's, will develop in a straight line towards clinical practice. This should really be dealt with in the discussion, the least by providing arguments for the persuasiveness of the suggested implementation scenario.,

Specific remarks:

-Classification statistics usually work with cut-off points/scores in a range. It is not explained how firm this cut-off is established whether there is a certain variation around the scores and whether these could vary or change based on larger series. As this possibly affects the calculations a scenario of further test development might be an option.

-If the test becomes popular with GP's, an increase of imaging and test costs, absolute numbers of false positives and false negatives may be the result. Was that scenario explored? The actual decision process on the scenario is, by the way, not described. Was there a scoring system used to establish the parameters in the scenarios?

-Does the literature on cancer screening support relevant background info for a test for triage use; that should be properly explained.

-the model used for CEA calculation is not explained -apart from the decision tree- and this should provide more detail; no methods description is provided in the abstract whatsoever.

- it seems the authors cannot think of any limitations in this

	paper/study as none are reported in the abstract. I fear that only after reviewing the response to these comments, I could actually advise the editor on acceptance.
REVIEWER	Randa Al Okka National Centre for Cancer Care and Research Doha - Qatar
REVIEW RETURNED	31-Oct-2017
GENERAL COMMENTS	Great manuscript

VERSION 1 – AUTHOR RESPONSE

Editor Comments to Author:

- The clinical expert group in the appendix - have these individuals given approval for their names to be published? If not, please remove this section.

The individuals listed have been instrumental in the development of the blood test and have been actively involved in the formation of the manuscript. All have given approval for their names to be included.

- The Strengths and Limitations section should be formatted as bullet points and should just outline the strengths and limitations of the study and study design. It should not serve as an article summary.

This has now been amended accordingly.

- We do not require the 'what is already known' 'what this study adds' box - please remove.

Noted – this has now been removed.

- Please discuss the limitations of the study in the discussion section.

Thank you – we have now extended our discussion of limitations in this study.

Reviewer(s)' Comments to Author:

Reviewer: 1

Reviewer Name: Giuseppe Bellisola

Institution and Country: Department Pathology and Diagnostics, Azienda Ospedaliera Universitaria Integrata Verona, Italy

Please state any competing interests or state 'None declared': None declared.

Based on the data obtained in a previous case-control study (Refs 2 and3), authors built models to explore the potential economic impact of adding a serum blood-based spectroscopic test in the triage for brain tumour diagnosis.

They aimed at achieving three main objectives: 1) define where the test could be used in the clinical pathway; 2) assess the potential cost-effectiveness of the technology; 3) define the level of performance necessary to meet the cost-effectiveness criteria of an health technology assessment (HTA) agency necessary to propose of extending trials on large cohorts of patients.

Parameters have been carefully selected as well as methodology rigorously applied to build predictive models and obtain health economic evaluation.

Results have been extensively illustrated and discussed. Check the correspondence with Figures number (Pages 28-29/40): ICER with varying test specificity is illustrated in Figures 5 and 6 and PSA in Figures 7 and 8, respectively.

The authors thank the reviewer for their careful consideration of the manuscript. The figure numbers have now been checked and updated.

Reviewer: 2

Reviewer Name: W.H. van Harten

Institution and Country: Netherlands Cancer Institute Amsterdam NL; University of Twente, Enschede NL

Please state any competing interests or state 'None declared': None Declared

This paper describes the Cost Effectiveness Analysis (CEA) of adding a protein spectrum test in serum to assess the chance of brain tumor presence in 2 scenarios.

The literature and background material on the test is not very extensive, so it is very difficult to assess whether the test is really specific for brain tumors (especially the hypothesis that the test covers all brain tumor types is surprising).

In our endeavour to focus upon the economic assessment of the test, we have reduced the clinical background information behind the test. We have expanded the background information where possible, but for conciseness, we have directed the readers to additional publications that can provide further information.

Further the background publication in J Neurooncol. suggests that the test also discerns other tumor types than brain tumors. I am no specialist in spectroscopy, but remember that early test series in my own institution, revealed very unspecific patterns, that could hardly discriminate between tumors, but also with a range of other diseases. The readers should be reassured in text that there is sufficient evidence that the test is specific for brain tumors and no other (non malignant) diseases are "hit" with this serum protein pattern.

We appreciate the reviewer taking the time to consider the background literature. The background publications cited are specific to research within the Spectral Analytical Laboratory at the University of Strathclyde, but there is also a number of proof-of-concept studies using this technique across a range of cancers and disease. Through our retrospective studies, the results indicate that we are specific and sensitive for brain tumours, and have an ability to differentiate between brain tumour origins too. This is associated with the highly dimensional data that is generated by ATR-FTIR spectroscopy, but also the powerful computational algorithms that are able to learn extremely specific disease signatures. It is our research into both of these aspects of the approach which have allowed this test to develop to the diagnostic performances illustrated. Our studies to date provide promising results, and our planned clinical based study on a prospective patient cohort will provide further data to substantiate this.

It is not described what the findings are of applying this test in a more general headache population. That is important as this may influence the methodological characteristics of the test and especially for the 1st scenario as GP's may send much more patients than expected once the test is commercialized.

The reviewer raises an important aspect of this study – if the test proves popular within the first clinical scenario, typically the general headache population, there is the possibility of overuse of the test resulting in higher demand on imaging. We make this comment within the text as this is a legitimate limitation of such an accessible test. Our focus is on the application of this test as a triage tool, where the blood test supplements a clinical decision-making tool helping to prioritise for urgent imaging those patients most likely to benefit from an early diagnosis. In our future clinical study, we will directly observe referrals from primary care, so that a clear picture of GP decision making can be obtained.

Further the authors seem not to be very convinced of the diagnostic value of the test, in view of not only presuming that 50% of the population that is at risk in General Practice, but also all referred patients will still undergo CT/MRI. It would be wise to explain the need for this more extensively, and especially the second scenario seems rather illogical, why adding a spectroscopy test while all patients go through imaging, which is necessary for proper diagnosis anyway. I would expect a scenario in which a proportion of referred pts scoring negatively on the test, does not need imaging. What is the rationale for adding the test in the present scenario?

The rationale behind the ‘50% presumption’ [that a negative test result will still result in referral and imaging], is that some presentations will require neuroimaging even if cancer has been ruled out because of the other diagnoses that are under consideration. For example, if a negative test result coincided with particular neurological features or worsening symptoms, a GP may be sufficiently concerned to still refer the patient and the diagnosis of stroke may be made. We have now added some further description to the text to highlight this reasoning. The ‘overestimate’ of the likely imaging rate following a negative blood test result underlines the likely added economic value of our test in real-life.

In the second scenario, the reviewer is right – all patients in this setting will still receive a brain scan. This decision has been made by the GP, as they were sufficiently concerned to refer this patient into secondary care. At this point the blood test is no longer a triage tool for GPs, but instead a triage tool for the radiologists who have a high demand on imaging equipment. This scenario describes the health economic benefits of prioritising the patients who are awaiting imaging. Evidence suggests average time-to-diagnosis for outpatient referrals of 28 days compared to 8 days for emergency cases (Aggarwal 2015) in England. Of all brain scans conducted, only 1 in 33 identifies the presence of a brain tumour. If that one patient can be prioritised and assessed more quickly than others in this pathway, this scenario describes the cost and health benefits that early diagnosis will have.

In early stages of test/marker development those directly involved can be rather optimistic about the value and seem often certain about the actual use in practice. After more thorough research this does not always hold and sometimes indications for application change and even competing tests can appear on which key opinion leaders’ perspectives may differ thus hampering system wide implementation. None of these options is explored in the discussion; I would consider it unlikely that the test on which only a few publications exist and is explored through CEA in a theoretical population in 2 scenarios will develop in a straight line towards clinical practice. This should really be dealt with in the discussion, the least by providing arguments for the persuasiveness of the suggested implementation scenario.

Undoubtedly, there is further clinical work required to validate this test and assess its use in clinical practice. There are currently two clinical studies in the pipeline; (i) a feasibility study in secondary care and (ii) a large trial in primary care. The former of these studies will be conducted alongside open access CT scanning facilities, and will allow us to determine the prevalence of disease and patient pick up rates. The latter will more closely mimic the tests implementation in primary care and will look across a larger cohort of patients. Prior to beginning this work, the authors believed that a health

economic assessment such as this was pivotal to understanding the clinical need for a test such as this; hence why this study has come before results of the aforementioned clinical studies. We believe this is beneficial to not only raise awareness for this test, but also to highlight the space for healthcare improvements specific for brain tumour diagnostics. We have now expanded the discussion and added comments raised by the reviewer. This now includes reference to the limitation that there may be new evidence and new technologies which are not foreseen in this early evaluation. And that there is therefore a need to update the analysis to inform future system wide implementation decisions at the time.

Specific remarks:

-Classification statistics usually work with cut-off points/scores in a range. It is not explained how firm this cut-off is established whether there is a certain variation around the scores and whether these could vary or change based on larger series. As this possibly affects the calculations a scenario of further test development might be an option.

Excellent point – thanks for raising this. The classification statistics quoted here are the average of 96 independent model iterations, specifically 96 iterations of a random forest algorithm. The standard deviation for the 92.8% and 91.5% sensitivity and specificity were respectively, 1.1% and 1.9%. This detail has now been added to the manuscript. The output of a random forest algorithm is typically a classification into positive and negative cases. However, this can also be interpreted as producing 'scores' that are predicted probabilities with a cut-off probability of 0.5 for being classified positive.

-If the test becomes popular with GP's, an increase of imaging and test costs, absolute numbers of false positives and false negatives may be the result. Was that scenario explored?

The potential for the test to be used in a broader population than we defined in scenario 1 was considered. This may lead to a 'spectrum bias' type effect with more false positives and false negatives in a less selected patient population. We considered a scenario with a 0.5% (base case) prevalence of disease as well as 1% prevalence of disease (appendix table 8). This demonstrates that lower prevalence leads to reduced cost-effectiveness and higher prevalence improved cost-effectiveness. Therefore, it would be expected that if prevalence was even lower than 0.5% in scenario 1 (for example an undifferentiated headache population) then the cost-effectiveness would be substantially reduced. The documented prevalence in the undifferentiated headache population is actually as low as 0.1% (Hamilton, W. & Kernick, D. Clinical features of primary brain tumours: a case-control study using electronic primary care records. *Brit J Gen Pract* 2007; 57: 695–9), and for this reason this scenario was not considered. Furthermore, the clinical expert group considered use in the broader population beyond those currently referred to direct access imaging was too uncertain. Planned future trials in primary care will carefully address the issue of patient selection and GP decision making to allow any updated economic evaluation to more rigorously define the relevant patient population. While this is certainly of interest, it is beyond the scope of this study to address.

- The actual decision process on the scenarion is, by the way, not described. Was there a scoring system used to establish the parameters in the scenarios?

The scenarios were defined by an iterative process between the project team and clinical expert group. The project team presented potential model scenarios and the expert group provided comments and suggestions. The scenarios were then modified and presented again to the expert groups until consensus was reached that plausible models had been constructed. The decision process for selecting patients to receive the test in the scenarios is not described in detail as it mimics existing practice for selecting patients for direct access imaging referral or secondary care imaging referral, in scenarios 1 and 2 respectively. Most model parameters were established by rapid reviews of the relevant clinical literature identifying the referenced sources. Those that were informed by

clinical expert opinion were based on achieving consensus in the clinical expert group on the parameter value.

-Does the literature on cancer screening support relevant background info for a test for triage use; that should be properly explained.

This is an interesting point. There are similar scenarios in the broader cancer screening literature. However, it should be noted that we consider only symptomatic populations while cancer screening programmes typically target asymptomatic populations. An example that is similar in nature to what is proposed in this study is the use of faecal immunochemical tests (FIT) for colorectal cancer to stratify patients when considering referral for colonoscopy (Auge, Josep M., et al. "Risk stratification for advanced colorectal neoplasia according to faecal haemoglobin concentration in a colorectal cancer screening program." *Gastroenterology* 147.3 (2014): 628-636.). A full discussion of these types of clinical scenario is probably beyond the scope of this study. A small addition has been made to the section mentioning cancer screening to clarify that this study only considers symptomatic patients. The introductory text relating to screening programmes is to provide the context in relation to brain cancers compared to other cancers.

-the model used for CEA calculation is not explained -apart from the decision tree- and this should provide more detail; no methods description is provided in the abstract whatsoever.

We were limited somewhat by the word limit in the text; however, further details of the CEA methods are found in Appendix 2 including all assumptions and parameters values which were used. One aspect that was missing was an explanation of the ICER calculation. This has been added to the methods section on page 13 and 14.

- it seems the authors cannot think of any limitations in this paper/study as none are reported in the abstract.

This is a much needed addition. The abstract has been restructured and now includes a strengths and limitations section. The key limitations of this study are now listed there.

I fear that only after reviewing the response to these comments, I could actually advise the editor on acceptance.

Thank you for taking time to review this manuscript and providing useful feedback. We hope that the responses sufficiently answer your questions.

Reviewer: 3

Reviewer Name: Randa Al Okka

Institution and Country: National Centre for Cancer Care and Research, Doha - Qatar

Please state any competing interests or state 'None declared': None.

Please leave your comments for the authors below Great manuscript

Thank you for your considering this manuscript.

We would again like to thank the reviewers for their valuable insight into our manuscript, and we have taken all feedback on board. We hope that any concerns have been adequately addressed by these edits and additions to the manuscript. This work is not under consideration elsewhere nor has it been previously published. We look forward to hearing back from you.

VERSION 2 – REVIEW

REVIEWER	Giuseppe Bellisola Department of Pathology and Diagnostics, Azienda Ospedaliera Univeristaria Integrata Verona, Italy.
REVIEW RETURNED	28-Dec-2017

GENERAL COMMENTS	Excellent work.
-----------------

REVIEWER	W.H. van Harten Netherlands Cancer Institute & University of Twente, The Netherlands
REVIEW RETURNED	11-Jan-2018

GENERAL COMMENTS	The manuscript is much improved following the earlier comments of various reviewers. I do however feel that the general tone is still rather optimistic, while the test is still in a very early development phase and actual larger prospective trials are still to be conducted. This concern does not very well appear in both the introduction and discussion parts and the fact that only two scenarios are been studied is also not stipulated very clearly; other scenarios are conceivable. If technical evidence is not as convincing as researchers now expect, if Key Opinion Leaders in the US are not easily convinced and actual diffusion is much slower, what does that mean for the calculations? Also the simple projection of UK figures on USA data (which exactly) is somewhat tricky. This should be very clearly specified as practical circumstances, defensive medicine, uptake without skipping other activities etc etc, are easily influencing the results. In all my advice is to incorporate these cautions much stronger in both the introduction and discussion parts than authors have done so far. As the core calculation part does not need further adaptations, I consider this minor but essential changes, hence my advice still to conduct minor revision.
---

VERSION 2 – AUTHOR RESPONSE

Reviewer(s)' Comments to Author:

Reviewer: 1

Reviewer Name: Giuseppe Bellisola

Institution and Country: Department Pathology and Diagnostics, Azienda Ospedaliera Universitaria Integrata Verona, Italy

Please state any competing interests or state 'None declared': None declared.

Excellent work

The authors thank the reviewer for their careful consideration of the manuscript.

Reviewer: 2

Reviewer Name: W.H. van Harten

Institution and Country: Netherlands Cancer Institute Amsterdam NL; University of Twente, Enschede NL

Please state any competing interests or state 'None declared': None Declared

The manuscript is much improved following the earlier comments of various reviewers. I do however feel that the general tone is still rather optimistic, while the test is still in a very early development phase and actual larger prospective trials are still to be conducted.

We appreciate that the tone of the article is a little optimistic. We have now incorporated the key points from the 'Strengths & Limitations' into both our Introduction and our Discussion. This includes highlighting that these results are based upon retrospective samples prior to a large-scale clinical trial, and the subsequent implications of this fact.

This concern does not very well appear in both the introduction and discussion parts and the fact that only two scenarios are been studied is also not stipulated very clearly; other scenarios are conceivable.

This is indeed true, and in the process of our study we considered a number of additional scenarios; including the possibility of using the test result as an absolute diagnostic, or also as a 'gatekeeping' tool in primary care, where a positive result would recommend referral to secondary care, but would not impact the timing of the imaging test. As you can imagine the possibilities are extensive and we tried here to focus on the most significant, and realistic, scenarios. We have added a comment in the discussion to address this.

If technical evidence is not as convincing as researchers now expect, if Key Opinion Leaders in the US are not easily convinced and actual diffusion is much slower, what does that mean for the calculations? Also the simple projection of UK figures on USA data (which exactly) is somewhat tricky. This should be very clearly specified as practical circumstances, defensive medicine, uptake without skipping other activities etc etc, are easily influencing the results.

Thank you for bringing this to our attention. Due to the inherent differences in the two healthcare systems, it should be expected that uptake may differ in the two localities. We have commented upon this in our Discussion to highlight this is an assumption made. At this stage of development, we feel that further exploration of the health economic calculations in the USA are beyond the scope of this particular article, which would require the results of clinical trials and cooperation with clinical experts in the USA.

In all my advice is to incorporate these cautions much stronger in both the introduction and discussion parts than authors have done so far. As the core calculation part does not need further adaptations, I consider this minor but essential changes, hence my advice still to conduct minor revision.

Many thanks to the reviewer for considering our article. We hope the additions above satisfy the reviewer and sufficiently cover some of the limitations and cautions of our study.

We would again like to thank the reviewers for their valuable insight into our manuscript, and we have taken all feedback on board. We hope that any concerns have been adequately addressed by these edits and additions to the manuscript. This work is not under consideration elsewhere nor has it been previously published. We look forward to hearing back from you.

VERSION 3 – REVIEW

REVIEWER	W.H. van Harten The Netherlands Cancer Institute; University of Twente. The Netherlands.
REVIEW RETURNED	11-Mar-2018

GENERAL COMMENTS	In all the authors have sufficiently addressed the concerns and remarks raised in the initial review. It is however a concern that the results are only based on pilot series and no real world testing has as yet been done (or is reported upon). It would not be the first time in which " inventors" are rather optimistic on the potential of a marker or screening test, whereas in real world scenario's may occur that are unforeseen seen, as raised in remarks in my initial review. Wider use among GP's than just for a subsection of cases, debate on acceptability among those not involved in the development or early test series on the underlying validity of data are just a few. These concerns should lead the authors to less firm statements in the discussion/conclusions setting and realism in the introduction about the scope of testing and results to be expected reasonably in this stage of development. These are minor but essential requirements.
--

VERSION 3 – AUTHOR RESPONSE

Reviewer(s)' Comments to Author:

Reviewer: 2

Reviewer Name: W.H. van Harten

Institution and Country: Netehrlands Cancer Institute Amsterdam NL; University of Twente, Enschede NL

Please state any competing interests or state 'None declared': None Declared

In all the authors have sufficiently addressed the concerns and remarks raised in the initial review.

Thank you, we are glad that you agree these concerns have been adequately addressed.

It is however a concern that the results are only based on pilot series and no real world testing has as yet been done (or is reported upon). It would not be the first time in which " inventors" are rather optimistic on the potential of a marker or screening test, whereas in real world scenario's may occur that are unforeseen seen, as raised in remarks in my initial review. Wider use among GP's than just for a subsection of cases, debate on acceptability among those not involved in the development or early test series on the underlying validity of data are just a few. These concerns should lead the authors to less firm statements in the discussion/conclusions setting and realism in the introduction about the scope of testing and results to be expected reasonably in this stage of development. These are minor but essential requirements.

We understand the concern expressed that this study may be misinterpreted as providing firm conclusions regarding cost-effectiveness of the technology which would indeed be inappropriate at this stage of development.

After reviewing the text, we have identified a number of places in which some additional caution in the language is appropriate. This includes several minor changes in the abstract, introduction and discussion sections. We believe these changes improve the consistency of the manuscript with the stated aim of the study to "assess the potential cost-effectiveness of this spectroscopic technology, in advance of any prospective study results being available" and the purpose of guiding the future

development of the technology and evidence base. We believe these changes address the concerns about this issue and ensure the study is correctly seen as a pre-trial evaluation with the limitations that entails.

Regarding the issue of unforeseen real-world scenarios in which the test is used differently to as intended in the examined scenarios, we aimed to analyse the potential cost-effectiveness of the technology in two scenarios defined by the clinical experts involved in the study. While many somewhat different scenarios are possible in the real world as you describe it is not possible to provide analysis for every possible case we could consider, and it impossible to do so for the truly unforeseen. This is related to the common criticism of randomised trial evidence as not being generalisable to the real world due to the fact that patients that tend to be included in randomised trials are different (selected in various ways) from those in the wider population that will receive the intervention in practice. This is a limitation of all randomised trials and all economic evaluations – it is not possible for a single trial or a single economic model to capture all aspects of real world practice. We believe this issue is addressed in the discussion section in which we highlight this issue of the need for future research studies using the technology to be responsive:

“Future developments beyond trials such as emerging epidemiological evidence and new technologies should also be included in any future evaluations. It was not possible to foresee and include all such possible scenarios in this early evaluation but that should not preclude assessment in the light of new evidence. Updated analysis should inform any decisions about system wide implementation.”

In order to further highlight this point, we have added an example in the manuscript which describes the potential for unexpected widening of the patient population considered suitable for the test. We note that this shows the need for the need to study decision making in this area prior to any implementation in primary care.

Our revisions have been highlighted in the attached manuscript.